# On the Calibration of Large Language Models and Alignment

**Chiwei Zhu[1], Benfeng Xu[1]\*, Quan Wang[2], Yongdong Zhang[1], Zhendong Mao[1]**

[1]University of Science and Technology of China, Hefei, China
[2]MOE Key Laboratory of Trustworthy Distributed Computing and Service,
Beijing University of Posts and Telecommunications, Beijing, China
`tanz@mail.ustc.edu.cn, benfeng@mail.ustc.edu.cn`
`wangquan@bupt.edu.cn, zhyd73@ustc.edu.cn, zdmao@ustc.edu.cn`

## Abstract

As large language models attract increasing attention and find widespread application, concurrent challenges of reliability also arise at the same time. Confidence calibration, an effective analysis method for gauging the reliability of deep models, serves as a crucial tool for assessing and improving their reliability. However, such investigation has been comparatively underexplored. In this work, we conduct a systematic examination of the calibration of aligned language models throughout the entire construction process, including pretraining and alignment training. At each stage, we investigate how different training settings, such as parameter scales and training data, affect model calibration. To thoroughly assess model calibration, we evaluate models on three most concerned aspects: generation, factuality and understanding. Our work sheds light on whether popular LLMs are well-calibrated and how the training process influences model calibration.

## 1 Introduction

Large Language Models (LLMs) like GPT-3 (Brown et al., 2020), PaLM (Chowdhery et al., 2022), and GPT4 (OpenAI, 2023), followed by many open-source replications including LLaMA (Touvron et al., 2023a), Pythia (Biderman et al., 2023) are revolutionizing the paradigm and re-shaping the expectation of modern natural language processing. When further trained with alignment treatment (Ouyang et al., 2022; Bai et al., 2022), these LLMs further exhibit impressive capability in responding to generalized human instructions, which implies their potential as general-purpose intelligent assistants and this has since attract considerable attention in the field and around the world.

As LLMs find more diverse applications and exert widespread influence, it becomes increasingly

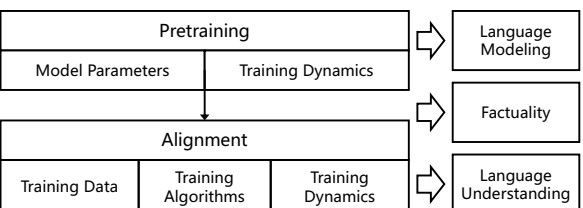

Figure 1: Scope of investigations in this paper.

imperative to ensure their reliability and faithfulness, particularly in fields such as healthcare (Kung et al., 2023) and law (Huang et al., 2023). These are domains where inaccurate predictions can lead to significant, potentially severe challenges. However, due to the intrinsic autoregressive mechanism and complex system structures, the behaviours of these models can not be easily attributed or interpreted.

Confidence calibration is an effective method to estimate a model's awareness of its uncertainty, and it helps enhance our understanding and assurance of the trustworthiness of deep models. Generally, it associates model output confidence, i.e. probability, with ground truth correctness likelihood (Guo et al., 2017) and informs the user to what extent the outputs should be trusted, even though they may not always be correct. Intuitively, for example, given 100 predictions in a classification task which are produced by a classifier and each of them is assigned 0.8 confidence, we expect 80 of them to be correctly classified for a well-calibrated classifier. As a consequence, better calibration of LLMs could significantly extend their usability. In early meteorology, calibration was noted as validity (Miller, 1962) or reliability (Murphy, 1973), indicating the trustworthiness of forecasters. Well calibrated probabilities can provide extra information for users to decide whether to trust the model's prediction, particularly for modern neural networks whose decisions are harder to interpret (Guo et al., 2017). Studies have also pointed out that calibration is helpful to reduce hallucination in language

---
\*Corresponding author: Benfeng Xu.

models (Xiao and Wang, 2021; Tian et al., 2020). Previous works have shown that pre-trained language models can generate well-calibrated predictions (Desai and Durrett, 2020; Kadavath et al., 2022). However, these works mainly concentrate on vanilla language models, while the aligned language models receive less focus. A newly proposed work evaluates calibration of some aligned models by prompting them to verbalize confidence in the token space (Tian et al., 2023), but it mainly studies black-box models, whose training process is not available, and thus can not provide insight into how model calibration is affected by different factors in the alignment training process. To conclude, a systematical study on the calibration of aligned language models is still missing, and our work aims to fill this gap.

In this work, we study the calibration of aligned language models in the entire building cycle and provide evidence on how to achieve decent model calibration. An overview of the scheme of out study is at Figure 1. Following the training process of aligned language models, we study model calibration in pre-training stage and alignment training stage respectively. In each stage, we reveal how model calibration changes when using different training settings. For pre-training stage, we examine the effect of parameter scale and training dynamics (steps). For alignment training stage, we study the effect of instruction tuning and RLHF, in which instruction tuning is further scrutinized by changing instruction datasets, training methods and also training dynamics.

Besides the understanding and generating ability, factual faithfulness and reasoning capability are two widely considered issues with large language models (Du et al., 2023). We also follow this path to study models' calibration when applied to different tasks. For this purpose, we design three tasks for each of the stages above. (1) To evaluate model calibration on common text generation, we use Causal Language Modeling (CLM) task, which is also the objective of pre-training stage. (2) To study model calibration on factuality, we designed a facts generation task where the models are asked to generate fact-related content. (3) To study model calibration on reasoning, we use multi-task language understanding task, where questions and possible options are provided and models are asked to select the most probable one.

Through extensive experiments and analysis, we arrive at the following findings.

**For pretraining of LLMs:**

- **Larger Parameter Scales** ↑ : **Improve** models' calibration.

- **Longer Training Dynamics** ↑ : Also **benefit** calibration accuracy.

**For alignment of LLMs:**

- **Instruction Tuning** ↓ : **Deteriorates** models' calibration.

- **Synthetic Data** ↓ : **Exacerbates** the harmful effect of instruction tuning.

- **Parameter-efficient Fine-tuning** → : Effective regularization for **restraining** calibration error.

- **RLHF** → : Help **maintaining** calibration accuracy.

**For different tasks:**

- **In pre-training:** Improvement in calibration accuracy is **more significant on fact generation task or language understanding tasks** than language modeling task.

- **In alignment training:** Calibration accuracy **evolves consistently across different downstream tasks** including fact generation, language understanding or vanilla language modeling.

We believe these conclusions as well as detailed experiments can take us a step further towards understanding large language models, especially the intrinsic mechanism of their calibration behaviour. Our experimental results also provide us with some possible solutions to improve calibration, including increasing model scales and employing parameter efficient tuning methods. Besides, diversity guided instruction data construction may also be very promising. Hopefully these findings can shed light on future works to construct more factual and trustworthy assistants.

## 2 Related Work

**Aligned Large Language Models** are large language models that are specially trained to follow human's intents or instructions. Large language

models are proved to have the ability of completing some downstream tasks without any gradient updating (Brown et al., 2020). To better make use of such ability, many researches have found that instruction following models can be constructed by fine-tuning models with instruction-response pairs, which is called instruction tuning (Weller et al., 2020; Mishra et al., 2022; Sanh et al., 2022; Wei et al., 2022a; Xu et al., 2023b). While these models can understand human instructions and make reasonable responses, they often produce unexpected results like lies, made-up facts, biased or toxic texts and so on. To better align models with human intents, reinforcement learning with human feedback is introduced to the training of large language models (Ouyang et al., 2022). Though instruction tuning and RLHF can significantly improve the models' ability of interacting with humans, how they influence the calibration of large language models have not been researched on.

**Confidence calibration** is a concerned problem for classification models. A large amount of works have studied the calibration of statistical machine learning systems and the methods to improve their calibration (DeGroot and Fienberg, 1983; Palmer et al., 2008; Yang and Thompson, 2010). Later, calibration of neural networks have also been researched on (Hendrycks and Gimpel, 2016; Nguyen and O'Connor, 2015; Nixon et al., 2019; Minderer et al., 2021). Guo et al. (2017) points out that modern neural networks are not as calibrated as their ancestors and proposes a temperature scaling methods to calibrate neural networks. In natural language processing field, calibration of transformer-based language models are evaluated among different tasks, including machine translation (Kumar and Sarawagi, 2019), QA (Jiang et al., 2021) and selective prediction (Varshney et al., 2022). Recently, large-scale generative language models are receiving growing attention, and some works have examined calibration of these models (Srivastava et al., 2022; Kuhn et al., 2023; Tian et al., 2023). There are also works improving calibration of large language models, for example, Xu et al. (2023a) propose $k$NN Prompting to effectively mitigate calibration errors in in-context learning. However, as mentioned before, these works either concentrate on vanilla language models or study black-box models. We study models calibration in their whole life cycles from pre-training to alignment training, where our main contributions

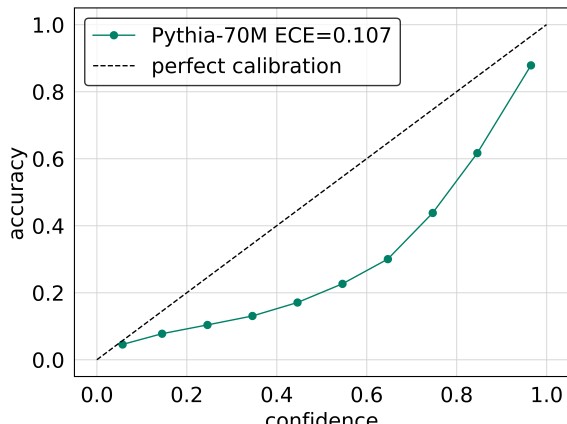

Figure 2: Reliability diagram for a Pythia-70m model.

lie in.

**Analysis of large language models.** Understanding various aspects of LLMs through theoretical or empirical approaches have long been an important interests for NLP scholars. Many works have demonstrated the scaling law of LLMs in different scenarios w.r.t. model scales, data size and computational costs (Kaplan et al., 2020; Rae et al., 2022; Xia et al., 2023). Wei et al. (2022b) defines and reveals the emergent abilities of large language models. Liang et al. (2022) proposes a holistic evaluation framework named HELM to analyze large language models on their capabilities, limitations, and potential risks. Hallucination and factuality also draw a lot of attention (McKenna et al., 2023; Zheng et al., 2023), but they do not take a further step towards the intrinsic mechanism while merely explore the verification on the surface. Differently, this paper provides a formal and systematical analysis on calibration behaviour of LLMs and their alignment treatment.

## 3 Definitions

In this section we formally define some basic concepts in our work, including confidence calibration, reliability diagram and expected calibration error. Confidence calibration is the main objective we are studying in this work and the other two are tools we use to evaluate model calibration. Our selection of tools follows previous work (Guo et al., 2017).

**Confidence Calibration.** Given a supervised multi-class classification scenario, where we have input $x$, label $y \in \mathcal{Y} = \{1, 2...K\}$, model prediction $y' \in \mathcal{Y}$ and confidence $p' \in [0, 1]$. A model is

perfectly calibrated, if we can get

$$P(y' = y | p' = p) = p, \quad \forall p \in [0, 1]$$

for any input $x$ (Guo et al., 2017). In another word, the more confident a model is, the chance of its prediction being the same as ground truth should be higher. It should be noted that $P(y' = y | p' = p)$ can not be calculated with finite number of samples, so calibration is often evaluated by some statisitcal approximations.

**Reliability Diagram** is a kind of visualized evaluation of confidence calibration (DeGroot and Fienberg, 1983), which plots prediction accuracy as a function of confidence (e.g. Figure 2).

To evaluate the calibration with a model with a finite set of samples, we divide confidence interval $[0, 1]$ into $M$ bins with equal length $(1/M)$ and group model predictions into these bins according to their prediction confidence. Let $B_m$ be the set of indices of samples which fall into the interval $(\frac{m-1}{M}, \frac{m}{M}]$, then for each interval bin, we can calculate corresponding accuracy and average confidence as follows:

$$Acc(B_m) = \frac{1}{|B_m|} \sum_{i \in B_m} \mathbb{1}(\hat{y}_i = y_i),$$

$$Conf(B_m) = \frac{1}{|B_m|} \sum_{i \in B_m} \hat{p}_i,$$

where $\hat{y}_i$ and $y_i$ are the prediction class and ground truth of the $i_{th}$ sample. $\mathbb{1}$ is the indicator function which produces 1 if $\hat{y}_i = y_i$ otherwise 0. $\hat{p}_i$ is the prediction confidence (probability) of the $i_{th}$ sample.

Given $Acc(B_m)$ and $Conf(B_m)$, we can draw the reliability diagram for a model. For a perfectly calibrated model, we will have $Acc(B_m) = Conf(B_m)$ for all $m$, so its reliability diagram will be $y = x$. Obviously, the nearer a curve is to the diagonal, the better calibration it represents. Though perfect calibration is impossible, we normally hope a model is well calibrated. Note that since reliability diagram do not contain the number of samples, sometimes it could be insufficient to represent true calibration of a model when some bins only contain very few samples.

**Expected Calibration Error (ECE).** As reliability diagram is more like a qualitative evaluation which depicts model calibration in different confidence intervals, we hope to get a quantitative scalar

metric which can reflect overall calibration level of a model. Expected Calibration Error is such a quantitative measurement of calibration (Naeini et al., 2015).

For a set of $N$ samples, we also divide confidence intervals into $M$ bins and get $Acc(B_m)$ and $Conf(B_m)$ in the same way as we did when drawing a reliability diagram. Then ECE is calculated as follows:

$$ECE = \sum_{m=1}^{M} \frac{|B_m|}{N} |Acc(B_m) - Conf(B_m)|$$

ECE represents confidence error averaged on samples and obviously lower ECE means better calibration. We set $m = 10$ when measuring calibration with the tools above following previous works (Guo et al., 2017; Desai and Durrett, 2020; He et al., 2023).

# 4 Calibration Evaluation Tasks and Data

As mentioned before, we evaluate model calibration on three tasks considering language models' ability of understanding, factuality and reasoning. In this section, we in detail introduce for each task how the evaluation is conducted and the datasets chosen for evaluation.

**Causal Language Modeling** is the task of predicting the next token for a given sequence, which is also the pre-training objective of causal language models. For a test sequence, we randomly sample a position in the sequence. Then this sequence is fed into models to generate a token corresonding to the position. If the predicted token is the same as the one in original sentence, we count it as a true positive. In such way we can get generation accuracy and confidence of test dataset, and then evaluate model calibration with reliability diagram and ECE metric. We use development and test set of the PILE dataset (Gao et al., 2020) in this task. The PILE dataset is a large-scale English text corpus which is frequently used in the pre-training of large language models.

**Facts Generation** is the task aimed at evaluating models' memory on factual knowledge. The task is mostly the same as causal language modeling in form , except that we utilize entity linking data, in which texts are labeled with entity spans. We let models generate the first token of entities. We only take the first token of entities into account as when the first token is correctly generated, there is high

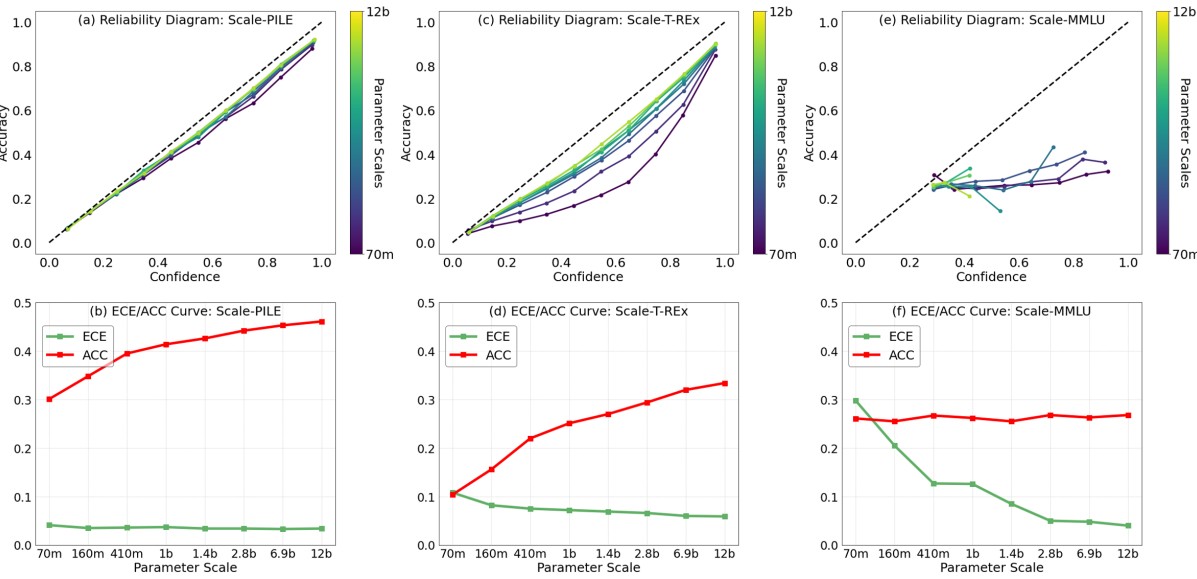

Figure 3: Model calibration of different parameter scales.

chance that the whole entity can also be recovered. In facts generation task, we use an enitity linking dataset T-REx (Elsahar et al., 2018), which includes entity-labeled texts extracted from Wikipedia pages. For T-REx and the PILE dataset, we randomly draw 100k samples as our evaluation set.

**Multi-task Language Understanding** is a task where models are given questions across different fields with multiple answer options, which is designed for testing the understanding and reasoning ability of a language model. We mainly focus on questions with a single correct answer. Following MMLU benchmark (Hendrycks et al., 2021), we concatenate 5 in-context samples ahead of the questions and designed prompts to constrain models to respond with answer options (i.e. 'ABCD'). We choose MMLU benchmark (Hendrycks et al., 2021) as our evaluation data, which covers single-choice questions in 57 subjects across STEM, the humanities, the social sciences and so on.

# 5 Calibration in Pre-training Stage

In this section, we study the effect of parameter scales and training dynamics in pre-training stage to models' calibration.

## 5.1 Experimental Setups

We choose Pythia as our base model (Biderman et al., 2023). Pythia is a suite of transformer-based, auto-regressive language models designed for scientific research. It contains 8 models whose scales range from 70m to 12B parameters and for each

of the scale it provides 154 checkpoints including 143 checkpoints saved every 1,000 training steps (i.e. 1 epoch) and 11 checkpoints trained for less than 1,000 steps. All of these models are trained on exactly the same data—the PILE (Gao et al., 2020) dataset in the same order. For parameter scale study, we experiment on models with all 8 scales. As for training dynamics, we choose Pythia-1B4 considering time and computational cost, and use $2^n * 1,000 (n = 1, 2...)$ steps checkpoints (up to step143,000) for our study. We also include checkpoints of step256 and step512 in our experiments to observe the behavior of under-fitted models.

## 5.2 Parameter Scales

Figure 3 shows the experimental results of parameter scales on three tasks. Generally, larger models produce better calibrated results while the level of such effect is diverse among tasks. We find that models with all parameter scales can produce well-calibrated predictions on CLM task, with ECE lower than 0.1. Also, parameter scales only mildly affect model calibration on the CLM task, where difference is minor between smallest and largest model. This might because CLM task is the same as the pre-training objective, where large scale and diverse corpus makes it hard for models to be over-confident when generating common texts. On facts generation task, model performance on both calibration and accuracy shows a stronger positive correlation with parameter scales. Results on MMLU (Figure 3-e and 3-f) seems very messy, but we can still observe some meaningful patterns here. All

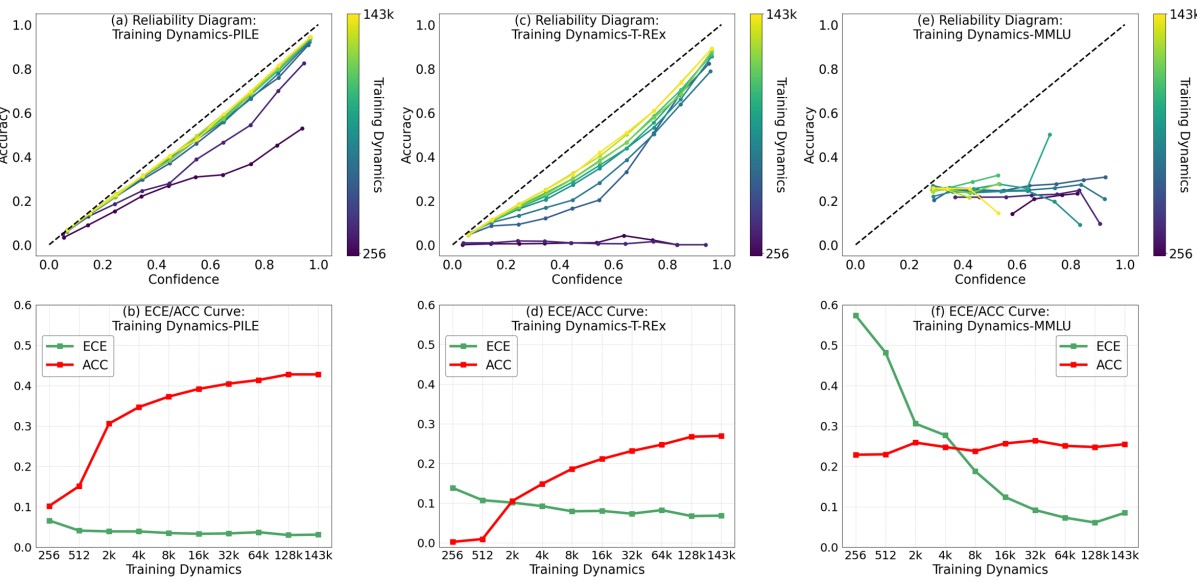

Figure 4: Model calibration of different training dynamics.

models perform poorly on language understanding task, with accuracies only slightly better than random choice. As Pythia models have not been trained to follow certain instructions, it is hard for them to understand these knowledge-demanding questions in the MMLU dataset. However, while accuracies from all models are similar, ECE decreases monotonically as the model scale increases. Moreover, we found that as the parameter scale increases, confidence distribution of model output gradually shrinks to a smaller and lower interval (see Appendix B), which might indicate that although larger models still can not solve these problems, they are more aware of their own capability than small ones. To further verify our conclusions, we conduct same experiments on 4 more models, LLaMA (Touvron et al., 2023a), LLaMA-2 (Touvron et al., 2023b), FLAN-T5 (Chung et al., 2022) and OPT (Zhang et al., 2022). Results are presented in Appendix C.1. We can see that our conclusions holds most of the time, where LLaMA2, FLAN-T5 and OPT show monotonic improvement with increasing model scale while there are outliers in the results of LLaMA. This demonstrates that factors other than scale may influence the trend, and LLMs should be examined using more continuously sampled checkpoints to reach a more robust conclusion, which we leave for future works.

### 5.3 Training Dynamics

On the whole, the effect of training dynamics follows the same pattern with that of parameter scales but there are a few unique observations can be

pointed out (See Figure 4). On CLM task, we can observe apparent improvement in both accuracy and calibration in the very early stage of pre-training. However, though accuracy keeps growing as the training goes on, ECE stabilize at a low level for the rest of the training process, which means that under-fitted models can also be well-calibrated. Training dynamics also show a stronger impact on facts generation task. It can be seen that models trained for less than 1 epoch behave extremely poor. In this stage, models are near to randomly initialized parameters which can not generate a reasonable probability distribution, thus the accuracy is almost zero for all confidence intervals. Results on MMLU dataset is similar to that of parameter scales, with accuracy barely grows while calibration keep improving. Note that we observe an increase of ECE in step143,000 (see Figure 4-f), which may be an sign of over-fitting. As Pythia does not provide checkpoints with further steps, we keep this as a simple hypothesis.

### 6 Calibration in Alignment Stage

Alignment training is divided into two sub-stages, instruction tuning and RLHF. In this section we study how model calibration changes during these process. As the models are fine-tuned on instructions in this stage, we add instruction prompt for all three tasks (see Table 1). Figure 5 shows the ECE level of fine-tuned models on three tasks when using different alignment training settings. We also report accuracy results in Appendix C.2, where the

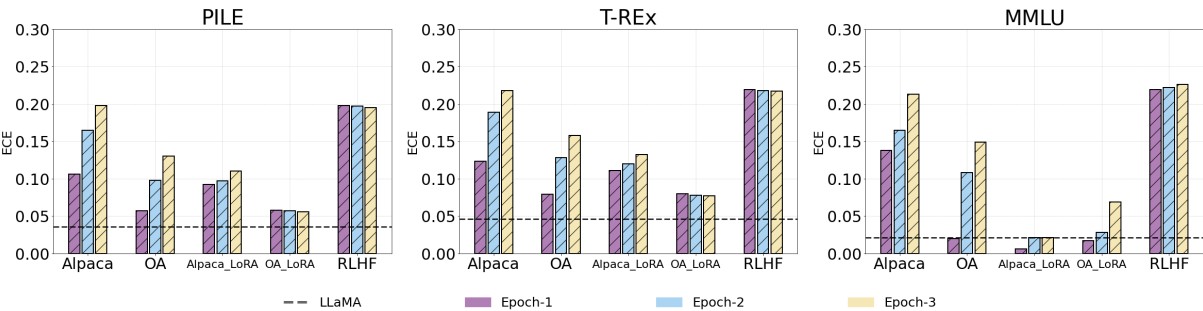

Figure 5: Model calibration using different alignment training settings.

| Tasks | Prompts |
|---|---|
| Causal Language Modeling | *Finish this sentence:* |
| Facts Generation | *Finish this Wikipedia description:* |
| Multi-task Language Understandign | *The following are multiple choice questions (with answers) about {...}.* |

Table 1: Prompts of different tasks in Alignment Training Stage

accuracy performance generally remains stable and only fluctuates with different datasets and training methods.

## 6.1 Experimental Setups

In alignment training stage, we use LLaMA-7B as our base model (Touvron et al., 2023a). Complete training hyper-parameters can be found in Appendix A.

**Instruction Tuning.** We leverage open-source GPT-generated and human-labeled data in instruction tuning. Alpaca (Taori et al., 2023) contains 52k pairs of instructions and responses generated in the style of self-instruct (Wang et al., 2022) using OpenAI GPT model. OpenAssistant Conversations[1], which we denote as OA, is a human-labeled assistant-style conversation corpus. We extract all single-turn conversations written in English from OA, resulting in 11k pairs of instructions. Considering fairness, we sample the datasets to the same size when comparing effect of instruction tuning. We follow setups of Stanford-Alpaca[2] and Alpaca-LoRA[3] for direct fine-tuning and LoRA training respectively. For each experiment group, we train the model for 3 epochs.

**RLHF** training contains two parts, reward model training and reinforcement learning. Training of reward models often needs ranked response data, where responses from different language models to the same instructions are collected and ranked by human annotators or another language model. We use open-source LM-ranked responses data (Peng et al., 2023), where GPT-4, GPT-3.5 and OPT-IML generate responses to Alpaca instructions and these responses are ranked by GPT-4. We use LLaMA as our reward model and conduct PPO training on top of previously instruction tuned model with Alpca data. We also use LoRA in RLHF training process to lower computational costs. We perform RLHF training with Huggingface TRL Library[4].

## 6.2 Instruction Tuning

We find that instruction tuning generally weakens the calibration of language models. Such impact changes when using different instruction tuning settings.

**Training Data.** As can be seen in Figure 5, direct fine-tuning with Alpaca does the most harm to calibration while models fine-tuned with OA dataset perform better. Note that in MMLU dataset, the model trained with OA is even better calibrated than LLaMA in the first epoch, which might because of the capability gain in following instructions. However, the ECE level rapidly becomes worse in later epochs, which represents that degeneration of calibration is a result of fitting process and OA is less likely to cause such degeneration. We presume such behavior is related to the diversity of datasets. Such presumption is mostly based on intuition where samples in OA datasets display strong personal or emotional characteristics while

---

[1]https://huggingface.co/datasets/OpenAssistant/oasst1
[2]https://github.com/tatsu-lab/stanford_alpaca
[3]https://github.com/tloen/alpaca-lora
[4]https://huggingface.co/docs/trl/index

instructions in Alpaca are much more homogeneous. To further look into the difference of Alpaca and OA, we compare semantic diversity of their responses. We use MPNET (Song et al., 2020) to extract sentence features of responses in both datasets and visualize these features with t-SNE (Van der Maaten and Hinton, 2008), see Figure 6. Results show that semantic features of OA dataset are more evenly distributed while those of Alpaca tend to be dense and clustered, which means the former is more diverse in semantics. We attribute this difference of diversity to how the datasets are constructed. Alpaca is a synthetic corpus generated in a self-instruct way, where a small set of human-written instruction data containing 175 seed tasks are fed into GPT-3 and augmented to 52k scale. In this case, Alpaca will contain a lot of instruction data with similar format and content for each seed task, and fine-tuning with such clustered dataset consequently leads to a worse calibration. On the other hand, OA is created by crowd-sourcing where thousands of volunteers are asked to submit their own instruction data, which makes the dataset more diverse in tasks and text styles, thus do less harm to model calibration.

**Training Methods.** Parameter efficient tuning is a type of training methods that keep the pre-trained weight unchanged and only train a small set of extra parameters. Although these methods are originally designed to train large models with lower resources, they may be able to improve calibration by reducing catastrophic forgetting (He et al., 2023). We compare calibration of models trained on Alpaca and OA datasets with full fine-tuning and LoRA tuning. Results exhibit that in all three tasks, model trained with LoRA performs better in calibration than those are directly fine-tuned. Besides, it can be noticed that deterioration in calibration is slight for model trained with LoRA (often in a level of 0.001) when training epochs increase, while the full fine-tuned model becomes visibly worse. Such observations proves that LoRA can mitigate the calibration degeneration in the instruction tuning process. In MMLU dataset we observe that behaviors of models trained on Alpaca with LoRA are similar to those trained with OA, where calibration improves compared to LLaMA in the first epoch. This may also indicate that LoRA is helpful in reduce the harmful effect of instruction tuning and improve model calibration.

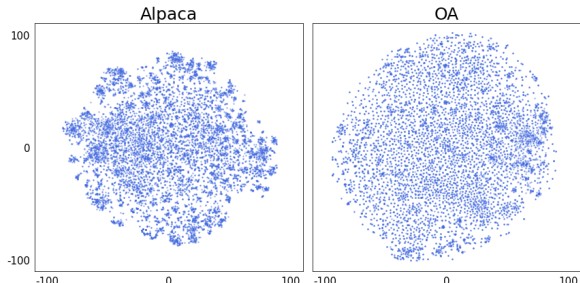

Figure 6: Sentence feature distributions of Alpaca and OA dataset (Each point is a response sentence).

**Training Dynamics.** In almost all instruction tuning experimental groups, models trained for more steps behave worse in calibration, which indicates that models calibration are affected severely by the small instruction dataset. Note that we obtained contradictory observation (model calibration improves when trained longer) in the pre-training stage where the model is trained with large-scale and diverse corpora, thus we anticipate that improve the scale and diversity of instruction data can also help improve calibration.

## 6.3 RLHF

The last groups of bars in each chart of Figure 5 show ECE level of models trained with RLHF for three epochs. We can see that compared to the instruction-tuned model, i.e. the 3rd epoch model trained with Alpaca, there is no significant degeneration in ECE after RLHF training. Moreover, models' calibration do not deteriorate as RLHF last for more epochs. This indicates that when applied to models that already have been trained with instruction data, RLHF might not do further harm to model calibration.

## 7 Discussion

Confidence calibration is helpful for building honest large language models in two ways. Firstly, confidence calibration is closely related to the uncertainty of language models, which is leveraged in many approaches like self-consistency to improve model performance (Wang et al., 2023). Studying and improving model calibration will provide further evidence to these methods and inspire more uncertainty-based techniques.

## 8 Conclusions

In this work we systematically study the calibration of aligned large language models. We designed

thorough experiments evaluating the model calibration with different training settings and reveal how model calibration is affected by pre-training and alignment training process. In pre-training, we find that model calibration improves as parameter scales and training dynamics increases. In alignment training stage, experimental results show that instruction tuning damages model calibration significantly and ill-distributed synthetic data does more harm. Such harm will increase when fine-tuning process lasts longer while can be remediated by using parameter efficient training methods like LoRA. In the mean time, we surprisingly find that RLHF has little impact on calibration of instruction tuned models.

## Limitations

There are two main limitations in this work. The first one is that we can not carry out fine-grained experiments on larger and better-performing models like LLaMA and LLaMA-2, as they only provide limited number of variants on scale (e.g. 7B/30B/70B for LLaMA-2) and do not provide checkpoints for different training dynamics. More detailed and rigorous conclusions can be drawn if finer-grained model variants are available. The second one is that our observations and conclusions can be further explored, like mining the relation of these observations to the mathematical theory of confidence calibration and proving our conclusions theoretically. We leave such in-depth exploration for future works.

## Acknowledgements

We would like to thank all the reviewers sincerely for their valuable advice to improve this work. This research is supported by National Science Fund for Excellent Young Scholars under Grant 62222212 and the General Program of National Natural Science Foundation of China under Grant 62376033.

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

# A Hyper-parameters

We list all used hyper-parameters in Table 2 and Table 3.

| Parameters | Values |
|---|---|
| **Direct Fine-tune** | |
| nums of gpu | 4 |
| epochs | 3 |
| batch size per gpu | 4 |
| gradient accumulation | 8 |
| total batch size | $4 * 4 * 8 = 128$ |
| max sequence length | 2048 |
| learning rate | 2e-5 |
| warmup ratio | 0.03 |
| lr scheduler | cosine |
| **LoRA** | |
| nums of gpu | 4 |
| epochs | 3 |
| batch size per gpu | 16 |
| gradient accumulation | 2 |
| total batch size | $4 * 16 * 2 = 128$ |
| max sequence length | 2048 |
| learning rate | 3e-4 |
| warmup steps | 100 |
| lora r | 8 |
| lora alpha | 32 |
| lora dropout | 0.1 |
| lora target modules | [q_proj, v_proj] |
| lr scheduler | linear |

Table 2: Hyper-parameters of instruction tuning.

| Parameters | Values |
|---|---|
| **Reward Model** | |
| nums of gpu | 4 |
| epochs | 2 |
| batch size per gpu | 8 |
| gradient accumulation | 1 |
| total batch size | $4 * 8 * 1 = 32$ |
| max sequence length | 2048 |
| learning rate | 2e-5 |
| lora r | 8 |
| lora alpha | 32 |
| lora dropout | 0.1 |
| lr scheduler | cosine |
| **RLHF** | |
| nums of gpu | 4 |
| epochs | 3 |
| batch size | 8 |
| gradientaccumulation | 8 |
| output max length | 128 |
| learning rate | 1.4e-5 |
| lora r | 16 |
| lora alpha | 32 |
| lora dropout | 0.05 |

Table 3: Hyper-parameters of RLHF.

## B MMLU Confidence Distribution

Figure 7 shows the confidence distribution of outputs of Pythia models from 70m to 12B. As is explained in Section 5.2, the ranges of the distributions tend to become smaller for larger models.

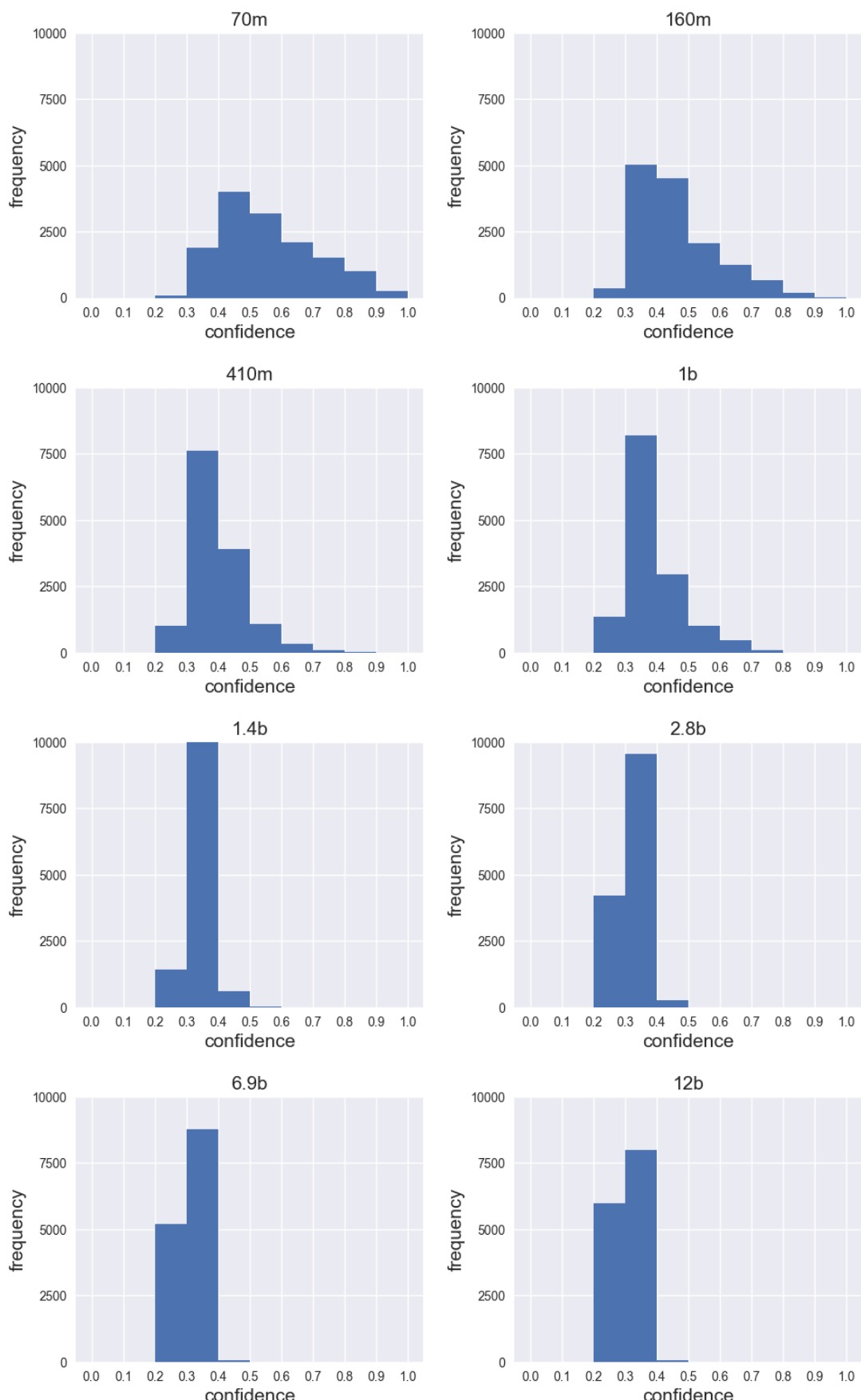

Figure 7: Confidence distribution of model outputs of different scales on MMLU Dataset.

# C  Supplementary Experimental results

## C.1  Calibration of Other Models

Figure 8-11 show supplementary results about parameter scales for other 4 models (LLaMA, LLaMA-2, FLAN-T5 and OPT), where we can draw similar conclusions with that of Section 5.2.

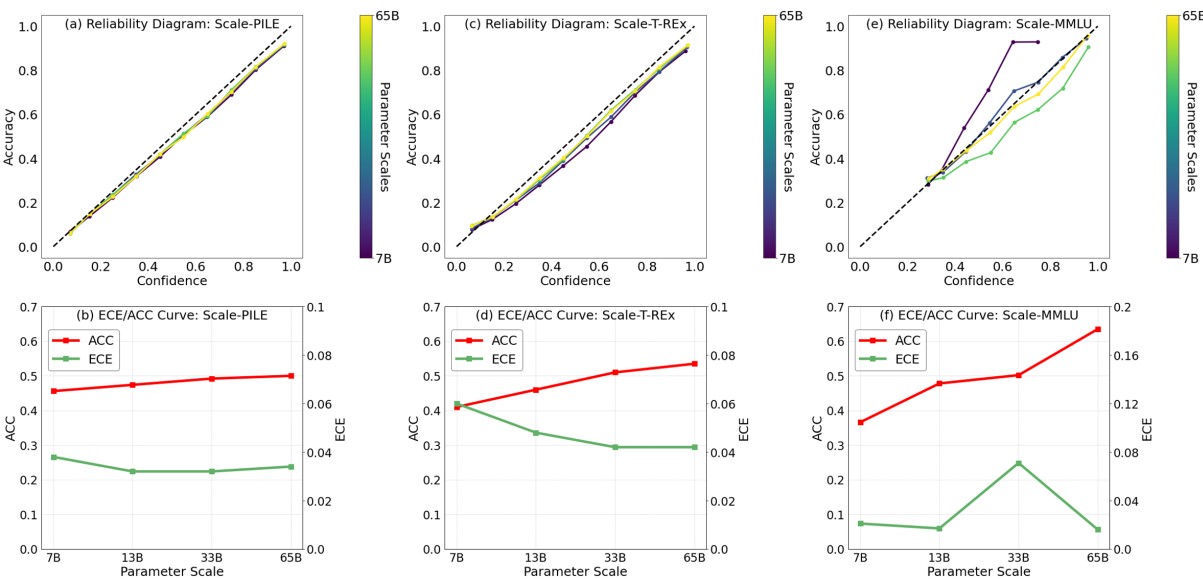

Figure 8: Calibration results of LLaMA.

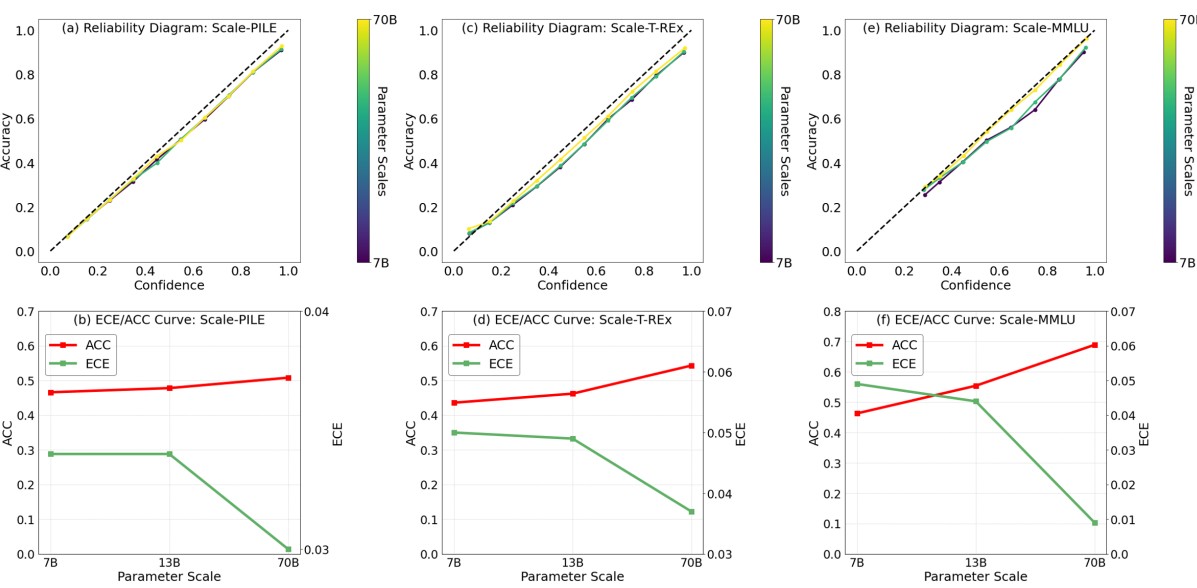

Figure 9: Calibration results of LLaMA-2.

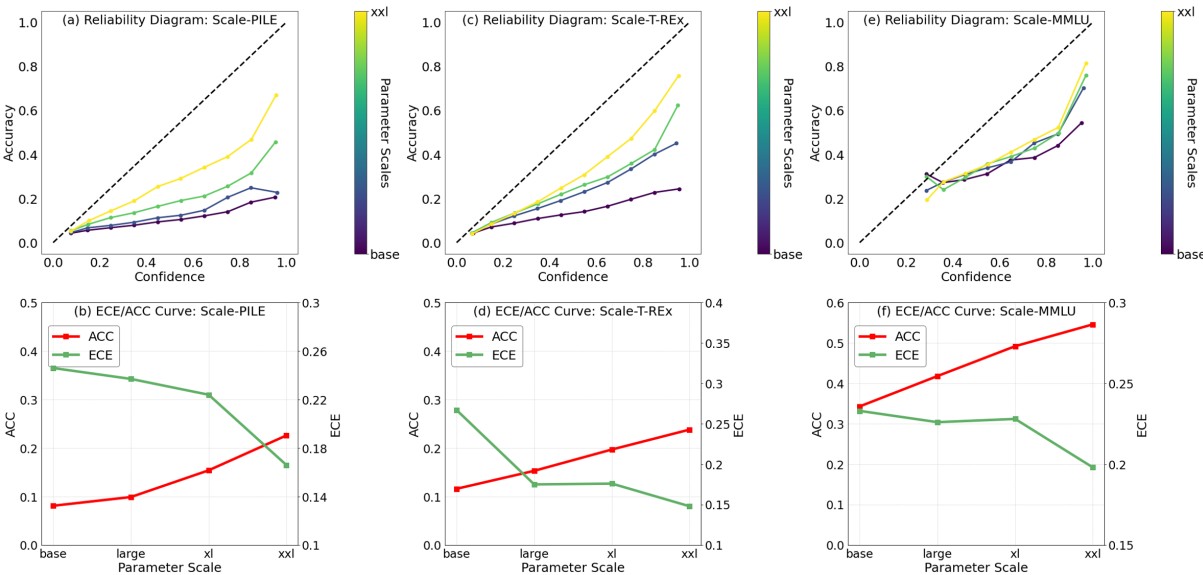

Figure 10: Calibration results of FLAN-T5.

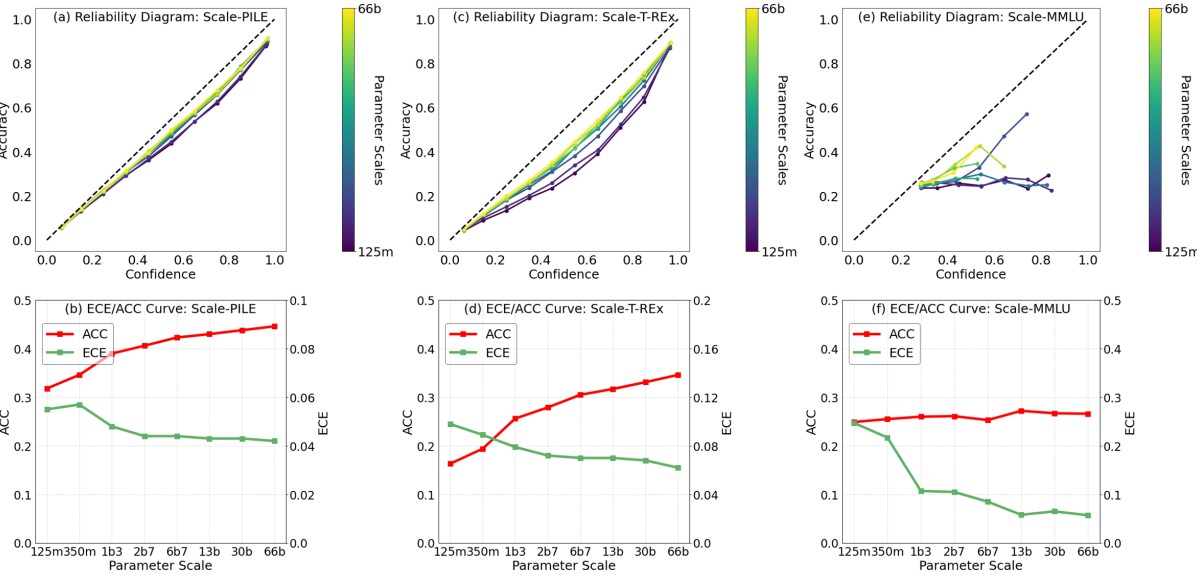

Figure 11: Calibration results of OPT.

## C.2 Accuracy Results in Alignment Stage

Figure 12 shows the accuracy of different model outputs on the the three datasets. As can be seen in the figure, accuracy generally keeps stable and only fluctuates with different training data.

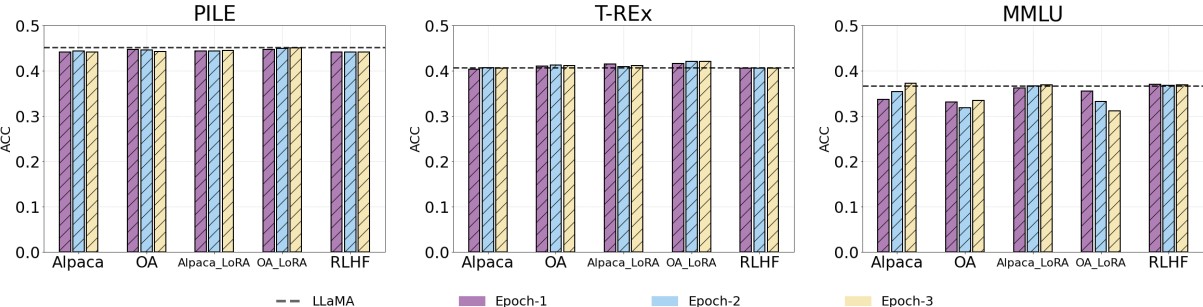

Figure 12: Model accuracy using different alignment training settings.