# OpenReview forum: "On the Calibration of Large Language Models and Alignment"
_EMNLP/2023/Conference — EMNLP 2023 Findings_

### Official Review · Reviewer_kGi5 · 2023-08-04

**Soundness:** 2

**Excitement:**

2: Mediocre: This paper makes marginal contributions (vs non-contemporaneous work), so I would rather not see it in the conference.

**Missing References:**

N/A

**Paper Topic And Main Contributions:**

Paper Topic
===
This paper studies the confidence calibration of large language models. They study how the calibration of LLMs changes with respect to the LLM size and the pre-training time on three tasks: causal language modeling, T-REx (factual knowledge), and MMLU (multitask language understanding). They also look into how instruction tuning and RLHF affect the calibration of LLMs. They find that large LLMs and LLM pre-trains for longer calibrates better. They also find that instruction tuning harms calibration ability while RLHF merely changes the calibration ability of LLMs.

Contributions
===
NLP engineering experiment

**Questions For The Authors:**

A. Is there any experiment results on OA_LoRA for Section 6.2?

**Reasons To Accept:**

1. Confidence calibration is vital for building and deploying safer and more helpful LLMs.
2. The paper is well-written and easy to understand

**Reasons To Reject:**

1. **Evaluation methods  are not very convincing**
    - Evaluating the calibration of causal language modeling (CLM) does not make sense to me. Unlike classification, which has a ground truth answer for a given question and allows one to calculate the accuracy, CLM does not have a ground truth token given a prefix. Given a prefix, the next token can be any token that can make the ongoing continuation a reasonable sentence. In this case, using the token in the original dataset is unreasonable. While prior works study the calibration for NMT, I argue that causal language modeling is different from translation as translation still has some ground truth for the target sentence given the source sentence. This is not the case for CLM. One may argue that when calculating perplexity, we also treat the sentences in the test set as ground truth. This is slightly different since, when calculating perplexity, we are not concerned about confidence and accuracy.
    - The evaluation for T-REx is also odd. First, I disagree with counting the model to predict accurately when it predicts the first token accurately. This is not how accuracy should be calculated. Next, even when the first token is not the expected token, it is still possible that the model completes the sentence in another way that answers the factual question accurately. Requiring the model to follow the predefined format for calculating the accuracy is too strict and might underestimate the accuracy.

2. **The performance on MMLU in Figure 3 and Figure 4 is too low to discuss the calibration**
    - The reason why we are interested in confidence calibration is that when we get a model that somewhat works, we want to use its confidence to tell us when it works and when it does not work. However, the performance of MMLU in Figure 3 and 4 is mostly as worse as random. In this case, it seems meaningless to discuss the models' confidence calibration since they are not usable.

3. **Section 6 (Instruction tuning and RLHF) does not show the accuracy/performance of models fine-tuned after instruction tuning and RLHF**
    - It is unclear if the instruction tuning and RLHF experiments in this paper really yield models that can follow instructions. Chances are the instruction tuning and RLHF is unsuccessful, and the loss diverges, so the tuned models cannot work well on any downstream tasks. Without any experiments testing the performance of instruction-tuned or RLHF-tuned models, it is hard to interpret the results here.

4. **Conclusions in this paper are drawn too dogmatically without enough evidence**
    - The connection between overfitting on instruction fine-tuning datasets and the ECE on the three tasks is not very clear. I do not see why the rising ECE for the three tasks during instruction tuning can be attributed to overfitting on the training dataset.
    - The connection between semantic diversity and ECE is not clear. The experiment in Figure 6 only shows a connection between semantic diversity (shown in T-SNE) and the ECE of the three test tasks. This is a correlation, not causation. OA and Alpaca may have differences other than semantic diversity. Without more analysis and careful interpretations, we can not know if semantic diversity is really the cause of lower/higher ECE.

5. **All experiments are based on a single run. It is unclear if the observations hold when changing the hyperparameter of fine-tuning the models**. RLHF is very unstable, and fine-tuning an LLM can have very different results depending on the hyperparameters, optimizers, and parameter-efficient fine-tuning methods used. Without more experiments, it is hard to tell if the results in this paper is a general trend or a random observation.

**Reproducibility:**

4: Could mostly reproduce the results, but there may be some variation because of sample variance or minor variations in their interpretation of the protocol or method.

**Reviewer Confidence:**

5: Positive that my evaluation is correct. I read the paper very carefully and I am very familiar with related work.

**Typos Grammar Style And Presentation Improvements:**

Line 054: is(?) should be trusted
Line 031: Missing reference
Line 192: Q A
Line 592: There is no OPT-130B
Line 218: `they do not take a further step towards the intrinsic mechanism while merely exploring the verification on the surface`. I feel like this paper also does not explore any intrinsic mechanism and only explores the superficial ECE, so it is hard to use the above sentence to justify the contribution of this paper.

---

> ### Author Rebuttal · Authors · 2023-08-29
>
> We appreciate the thorough and detailed review. Following are our replies containing further illustration and experimental results to hopefully address the reviewer's concerns.
>
> ### 1. The qualification of CLM and Fact Generation for calibration analysis
>
> > Evaluating the calibration of causal language modeling (CLM) does not make sense to me.
> >
> > The evaluation for T-REx is also odd.
>
> - For CLM task, if we look at a specific instance, it indeed “Not having a ground truth label”. But for a generalized distributed subset, the predicting accuracy along with the calibration error can well reflects the capability of LLMs and serves as sufficient metrics. For example, GPT4 ([1](#[1])) employs the accuracy on pretraining loss to predictably measure the overall capability of the model. On the other hand, In terms of the selected dataset, **PILE** contains a very **generalized distribution** of **high-quality** natural texts, including web corpus, academic papers, etc. The overall distribution of such aggregated natural texts can well represent how people use language, thus serves as qualified measurement for model capability as well as calibration behavior.
> - Note that for fact generation task, instead of directly ask factual questions, we performed evaluation in a CLM way where prefixes were given and models were asked to generate the entity token. Thus the problem about fact generation task can actually be induced to the first problem -- whether evaluation in the form of CLM is reasonable for model calibration, which we have illustrated above.
>
> [1] OpenAI (2023). GPT-4 Technical Report. *ArXiv, abs/2303.08774*
>
> ### 2. On the performance on MMLU in Figure 3 and Figure 4
>
> - First, regardless of the performance, we can already observe strong and clear patterns in the ECE results, calibration error **monotonically decreases** w.r.t. increased model scales and training dynamics, which can not be treated as randomness.
> - To further verify that such conclusions are not subjected to specific accuracy, we also apply the experiments of model scales on better-performing models(LLaMA). As can be seen in the following results(where the two tables represent accuracy and ECE respectively), with more accurate LLMs, the conclusions that larger models are more calibrated remain consistent. We will add these results in a revised version to make the arguments and analysis more solid. As for training dynamics experiments, we can not conduct further experiments for now as Pythia does not provide models with better performance on MMLU.
>
>     **Accuracy Results of LLaMA**
>
>     |       | LLaMA-7B | LLaMA-13B | LLaMA-30B | LLaMA-65B |
>     | :---: | :------: | :-------: | :-------: | :-------: |
>     | PILE  |  0.456   |   0.474   |   0.492   |    0.5    |
>     | T-REx |   0.41   |   0.46    |   0.51    |   0.535   |
>     | MMLU  |  0.367   |   0.478   |   0.579   |   0.635   |
>
>     **ECE  Results of LLaMA**
>
>     |       | LLaMA-7B | LLaMA-13B | LLaMA-30B | LLaMA-65B |
>     | :---: | :------: | :-------: | :-------: | :-------: |
>     | PILE  |  0.038   |   0.032   |   0.032   |   0.034   |
>     | T-REx |   0.06   |   0.048   |   0.042   |   0.042   |
>     | MMLU  |  0.021   |   0.017   |   0.025   |   0.016   |
>
> ### 3. Missing accuracy results on alignment
>
> - For the question that accuracy/performance in the alignment training is not provided, we apologize for the missing information. We will add the results in the revised version.
>
> - Accuracy results are as follows (Results are splitted into two tables for better view), in which the *LLaMA* column refers to results of the original *LLaMA* model, column labeled with *Alpaca/OA/Alpaca_LoRA* refers to performance of instruction-tuned models with different dataset or methods, and the last three columns are results of RLHF-tuned models.  From the table we can see that accuracy generally remains stable and only fluctuates with different datasets and training methods.  We also want to emphasize that this work mainly focuses on model calibration, and accuracy is just for reference.
>
>     |       | LLaMA | Alpaca_ep1 | Alpaca_ep2 | Alpaca_ep3 | OA_ep1 | OA_ep2 | OA_ep3 |
>     | :---: | :---: | :--------: | :--------: | :--------: | :----: | :----: | :----: |
>     | PILE  | 0.452 |   0.441    |   0.443    |   0.441    | 0.447  | 0.446  | 0.442  |
>     | T-REx | 0.407 |   0.403    |   0.407    |   0.406    |  0.41  | 0.412  | 0.411  |
>     | MMLU  | 0.366 |    0.336    |   0.354    |   0.372    | 0.331  | 0.318  | 0.334  |
>
>     |       | Alpaca_LoRA_ep1 | Alpaca_LoRA_ep2 | Alpaca_LoRA_ep3 | Alpaca_RLHF_ep1 | Alpaca_RLHF_ep2 | Alpaca_RLHF_ep3 |
>     | :---: | :-------------: | :-------------: | :-------------: | :-------------: | :-------------: | :-------------: |
>     | PILE  |      0.444      |      0.444      |      0.445      |      0.441      |      0.441      |      0.441      |
>     | T-REx |      0.415      |      0.409      |      0.411      |      0.405      |      0.405      |      0.405      |
>     | MMLU  |      0.362      |      0.367      |      0.369      |      0.37       |      0.368      |      0.369      |
>
> ### 4. Conclusions in this paper are drawn too dogmatically without enough evidence
>
> - Thanks for pointing out the problem of the word overfitting here. “Over-fitting” is indeed not a very rigorous description. The main observation in the paper is that “degraded ECE is clearly associated with training epochs”, longer training (i.e., fitting instead of over-fitting) consistently results in worse ECE error. In strict terms, training on the same dataset for more than one epoch does not necessarily results in overfitting (which is commonly recognized as when test errors starting to get worse while train errors keeps getting better).  We will edit this part in the revised version.
>
> - We agree that observations about diversity at present is correlation but not causation. The discussions in our paper are based on intuition at present. As can be seen in the cases about email below, instructions in OA show more personal characteristics while ones in Alpaca are more homogeneous. To strictly prove the relation of diversity and calibration, more quantative experiments should be conducted. We will re-form our conjectures to make it more rigorous and report any further experimental results in the revised version.
>
>     ```python
>     # OA
>     [
>         "I do not want to work today. Can you imagine a disease which makes me sick for three days and formulate an email for my boss that I cannot work for the next days?",
>         "I'm completely overwhelmed with all the work I need to do. So many people expect stuff form me and I just can't handle it all. The more I work, the less I get done and I have so many unanswered email and things I promised to do, I'm scared to even go look at my inbox. What can I do?",
>         "write an email for me to a journal that i submitted a paper in it, but the review process is taking too long and the review tracking webpage did not change in a long time. I want to ask them what's wrong or to speed the process up."
>     ]
>
>     # Alpaca
>     [
>         "Create a response for a customer's email about a refund issue.",
>         "Generate an example of a formal email.",
>         "Write an email introducing yourself to a professor you will be taking an online course with."
>     ]
>     ```
>
> ### 5. All experiments are based on a single run.
>
> - First, to prove that our observation is not a randomness, we report results of 5 runs on MMLU dataset using models trained on Alpaca with different random seeds (each cell contains results of three epochs). It appears that conclusions are the same for all runs and also the same with our conclusion in the paper (ECE increases monotonically with the training process). We will include results on other two datasets in a revised version.
>
>     **Accuracy and ECE results of 5 runs on MMLU dataset**
>
>     |          |       Run 1       |       Run2        |       Run3        |       Run4        |       Run5       |
>     | :------: | :---------------: | :---------------: | :---------------: | :---------------: | :--------------: |
>     | Accuracy | 0.366/0.324/0.336 | 0.385/0.377/0.388 | 0.336/0.369/0.38  | 0.379/0.348/0.367 | 0.36/0.396/0.388 |
>     |   ECE    | 0.019/0.085/0.125 | 0.038/0.035/0.07  | 0.066/0.084/0.132 | 0.023/0.137/0.174 | 0.03/0.117/0.165 |
>
> - Second, the main target of this work is to display the possible effect that pre-training and alignment training will bring to model calibration, instead of studying the influence of each hyperparameter. So we chose the training settings to change according to the design space of this work (Figure 1) and leave the others as commonly used or recommended.
>
> - Nevertheless, it is reasonable to experiment on more hyperparameters combinations to provide more solid conclusions. We will try our best to include these results in a revised version.
>
> ### 6. Experimental Results on OA_LoRA
>
> > Is there any experiment results on OA_LoRA for Section 6.2?
>
> - The reviewer mentioned that experiment results for OA_LoRA should be provided. Following are the results, from which we can see that **ECE increases in a much slower pace than models directly trained** (results in Figure 5). Such observation proves that our conclusion in previous experiments still holds (LoRA can mitigate the calibration degeneration caused by instruction tuning). We will add these results in the revised version.
>
>     **Accuracy Result of OA_LoRA**
>
>     |       | Epoch 1 | Epoch 2 | Epoch 3 |
>     | :---: | :-----: | :-----: | :-----: |
>     | PILE  |  0.447  |  0.449  |  0.45   |
>     | T-REx |  0.416  |  0.42   |  0.42   |
>     | MMLU  |  0.355  |  0.332  |  0.311  |
>
>     **ECE Results of OA_LoRA**
>
>     |       | Epoch 1 | Epoch 2 | Epoch 3 |
>     | :---: | :-----: | :-----: | :-----: |
>     | PILE  |  0.058  |  0.057  |  0.056  |
>     | T-REx |  0.08   |  0.078  |  0.077  |
>     | MMLU  |  0.017  |  0.028  |  0.069  |

---

### Official Review · Reviewer_ojDc · 2023-08-04

**Soundness:** 4

**Excitement:**

4: Strong: This paper deepens the understanding of some phenomenon or lowers the barriers to an existing research direction.

**Paper Topic And Main Contributions:**

This paper formally studies the impact of pre-training, instruction-tuning, and RLHF dynamics on the downstream calibration of model outputs across 3 tasks: causal language modeling (dev / test set of PILE), generating facts (T-REx), and NLU (MMLU benchmark).

Among others, they derive a few new insights on the impact of dynamics on calibration:
- calibration improves with model size and training times
- gains occur more dramatically for T-REx and MMLU benchmarks
- instruction tuning leads to worse calibration yet PEFT and RLHF do not impact it as much

**Reasons To Accept:**

Calibration is an understudied problem and, as the authors state, is closely linked to factuality (hallucinations) and trustworthiness of systems. They conduct a thorough investigation of each stage of the LLM pipeline from pre-training to instruction tuning to RLHF. They nicely include definitions and related works to help guide readers more unfamiliar with calibration literature.

**Reasons To Reject:**

The models tested only go up to 7B, and beyond 7B models behave much differently. While understandably difficult to finetune these larger models, it is possible to use them for inference on smaller GPUs. I think it's essential to add lessons from larger models in, even if the comparison is not apples to apples with the Pythia models. Showing Llama, Falcon, etc. calibration scaling laws would be helpful (for the model sizes -- the training length is not possible to compare).

Analogously, it's hard to read too much into the results because it's only tested on 3 datasets, and the CLM one does not feel particularly useful.  More effort needs to be done to motivate the choice of benchmarks and possibly why more benchmarks were not chosen.  Given that no benchmark specific tuning is necessary, hopefully this isn't too much effort.

Lastly, some of the text in the first pages could be tightened to allow for more of a deeper dive into the consequences of poorly calibrated models and possible ways to address it. Instruction tuning is a necessary step - so what should we conclude from this finding?

**Reproducibility:**

4: Could mostly reproduce the results, but there may be some variation because of sample variance or minor variations in their interpretation of the protocol or method.

**Reviewer Confidence:**

4: Quite sure. I tried to check the important points carefully. It's unlikely, though conceivable, that I missed something that should affect my ratings.

---

> ### Author Rebuttal · Authors · 2023-08-29
>
> Thanks for the valuable review and affirmative comments. As for some of the reviewer's concerns, we present our replies and supplemental results here:
>
> ### **Experimental Results on Larger Models**
>
> - Actually in the pre-training stage we tested models up to 12B (sorry for the mis-labeled legend in figure 3&4, we will correct them in the revised version). We have further conducted some of our experiments on larger models up to 65B (LLaMA). Results are as follows, the first table contains accuracy results and the second represents ECE. **For larger models, the conclusions made in the paper invariantly hold.** We can see that larger models possess better accuracy and calibration. While the accuracy improves significantly, the difference of ECE is minor as these large models are fairly well-calibrated in our three tasks and it might be difficult to improve calibration further. We will add these results in a revised version.
>
>     **Accuracy Results of LLaMA**
>
>     |       | LLaMA-7B | LLaMA-13B | LLaMA-30B | LLaMA-65B |
>     | :---: | :------: | :-------: | :-------: | :-------: |
>     | PILE  |  0.456   |   0.474   |   0.492   |    0.5    |
>     | T-REx |   0.41   |   0.46    |   0.51    |   0.535   |
>     | MMLU  |  0.367   |   0.478   |   0.579   |   0.635   |
>
>     **ECE  Results of LLaMA**
>
>     |       | LLaMA-7B | LLaMA-13B | LLaMA-30B | LLaMA-65B |
>     | :---: | :------: | :-------: | :-------: | :-------: |
>     | PILE  |  0.038   |   0.032   |   0.032   |   0.034   |
>     | T-REx |   0.06   |   0.048   |   0.042   |   0.042   |
>     | MMLU  |  0.021   |   0.017   |   0.025   |   0.016   |
>
>
> ### **On the choice of Benchmarks**
>
> - The motivation of choosing these datasets is that they are very representative, respectively for factuality (T-REx), general language understanding (MMLU), and text generation (PILE).
> - PILE itself is a very generalized high-quality texts, including Wikipedia, web corpus, academic papers or even code. It can already serves as qualified testbed for evaluating the essential calibration behavior of LLMs across various domains. We will consider other available datasets in future works.
>
>
> ### **About Writing**
>
> - Thanks for the insightful suggestion to include more introduction about the consequences of bad calibration and the addressing methods. In this work, we have already demonstrated that increasing model scales and employing parameter efficient tuning methods can effectively alleviate the degeneration of calibration. The experimental analysis also implies that **diversity guided instruction data construction** may also be very promising. We will add a detailed discussion in a separate section to shed more lights on effective or promising **solutions** for improving calibration in a revised version.

---

### Official Review · Reviewer_fbVH · 2023-08-04

**Soundness:** 3

**Excitement:**

4: Strong: This paper deepens the understanding of some phenomenon or lowers the barriers to an existing research direction.

**Missing References:**

The following reference is proposed if appropriate:

Lin, S., Hilton, J., & Evans, O. (2022). Teaching models to express their uncertainty in words. arXiv preprint arXiv:2205.14334.

**Paper Topic And Main Contributions:**

The paper focuses on the problem of calibrating large language models. This is a problem that has already been raised in the literature. The paper proposes an analysis covering the different training phases of current LLMs: a) Pre-training, b) instruction fine-tuning, and c) RLHF (phases b and c correspond to the process of alignment).

The calibration analysis is performed on three tasks that in some way represent the model's capabilities for language generation, fact generation and reasoning (underderstanding). The tasks are: Causal Language Modeling (PILE dataset), Facts Generation (T-Rex dataset), and Multi-task Language Understanding or multiple choice question (MMLU benchmark).

Reliability Diagram and Expected Calibration Error (ECE), already defined and used by Guo et al. (2017), are used as metrics to analyse the calibration of the models.

In the analysis of calibration in the pre-training phase, the effects produced by the size of the model (parameters) and the number of training steps are studied. For this analysis, the models of the Pythia suite, already created for different sizes and numbers of steps, are used. The results point to better calibration (and also accuracy) as the model size and number of steps increase.

In the analysis of the alignment phase, LlaMa-7B is used as a model. As datasets for instruction tuning, the use of Alpaca (synthetic) and OpenAssistant (manual) are analysed. For fine-tuning the model, the full fine-tuning and LoRA methods are compared. The results show that the calibration of the model degrades the higher the number of epochs of the fine-tuning process. This degradation is greater with Alpaca, and LoRA produces more calibrated models than full fine-tuning. The RLHF is performed on the Alpaca fine-tuned model using the data from Pen et al. (2023). It is observed that the RLHF hardly degrades the calibration of the model. However, the corresponding accuracyes are not presented in order to contextualise these calibrations.

Main contributions:

    - Analysis on factors influencing the calibration of LLMs throughout their construction (pre-training, fine-tuning and RLHF).

According to the results:

    - Calibration improves as the model size and amount of training increase.

    - Fine-tuning tends to deteriorate calibration, although this can be mitigated by using LoRA.

    - RLHF slightly deteriorates model calibration.

**Questions For The Authors:**

A) Does it refer to a understanding or generation?

[110] To evaluate model calibration on common text understanding, we use Causal Language Modeling (CLM) task,

B) What were the criteria for selecting Reliability Diagram and ECE for the analysis? It seems that the reference was the work of Guo el al. (2017). If so, this should be indicated.

C) The T-Rex dataset is mentioned to assess the Facts Generation task. However, PILE is also said to be used as a test (line 305). For what purpose is PILE used?

D) In line 406 it says "As the models are fine-tuned on instructions in this stage, we add instruction prompt for all three tasks (see Table 1)". I understand that the models are fine-tuned only on Alpaca and OA. Are these prompts for running the inference on the tests?

E) In the experiments on the alignment phase, why is LlaMA-7B used and not a Pythia model of those used in the pre-training analysis (e.g. Pythia-1B)? In that way, the evolution of accuracies and calibration along the different steps (pre-training, fine-tuning, and RLHF) could be analysed in a more coherent way.

F) Regarding the comparison between Alpaca and OA datasets, it is pointed out that OA implies more calibrated models because of its greater diversity. In addition to ECE, do you have results from any other metrics that point to this fact?
On the other hand, it is said that OA is less diverse because it starts from 175 seeds. And in one of the findings (line 132) it is said that "Instruction tuning deteriorates calibration, and the incorporation of synthetic data exacerbates such effect”. It is therefore assumed that all automatic datasets are inherently non-diverse. Is it safe to say that the automatic methodology determines less diversity in instructions than manual methods?

G) Why have accuracies not been included in the alignment experiments? They would be of great value in order to contextualise and interpret the ECE values in a more nuanced way.

**Reasons To Accept:**

• The main problem (calibration of LLMs) raised is important, and has not been extensively addressed in the literature.
    • Many of the specific aspects discussed in the paper are also unexplored in previous studies.
    • The proposed experimentation makes it possible to answer, albeit partially, the questions proposed in the analysis.

**Reasons To Reject:**

• There is no considerable contribution regarding SOTA.
        ◦  Kadavath et al., (2022) already point to a positive correlation between the calibration of LLMs and their size according to evaluations on tasks of diverse multiple choice and true/false questions. The contribution of this work seems limited to analyzing the number of pre-training steps and different configurations of fine-tuning and RLHF, in addition to including CLM and fact generation tasks in the analysis.
    • Some aspects of the experimentation and argumentation are weak, which prevent drawing solid conclusions.
    • The quality of the paper's writing could be improved in terms of spelling, grammar, and style.

**Reproducibility:**

4: Could mostly reproduce the results, but there may be some variation because of sample variance or minor variations in their interpretation of the protocol or method.

**Reviewer Confidence:**

4: Quite sure. I tried to check the important points carefully. It's unlikely, though conceivable, that I missed something that should affect my ratings.

**Typos Grammar Style And Presentation Improvements:**

The paper has much room for improvement in terms of writing. The following is an exhaustive list for the introduction only (the other sections require similar improvements).


031: A reference is missing.

032: capability at -> capability in

035: and has -> and this has

040: The paper (Kung et al., 2023) does not address reliability and trustworthiness.

043: potentially severe challenges. -> potentially severe, challenges.

046: are not easily attributed or interpreted.->are not easily attributable or interpretable.

053: (Guo et al., 2017). No earlier reference?

053: and inform -> and informs

054: outputs is should ->  outputs should

055: even though not always correct. ->even though they may not always be correct.

059: them are -> them to be

064: which indicates -> indicating

067: model prediction. -> model’s prediction

069: Studies also pointed -> Studies have also pointed

076: while the aligned language models are less focused on. -> while the aligned language models receive less focus.

082: thus can not provide insight on how -> thus can not provide insight into how

086: and our work is aimed to fill this blank. -> our work aims to fill this gap.

090: provide evidence on how to acquire ->  provide evidence on how to achieve

096: how model calibration is changed ->  how model calibration changes

099: For alignment -> For the alignment

106: are two widely concerned problems of -> are two widely considered issues with

108: “when applied to different fields.” It is not understood.

109: “For this purpose, we design three tasks for each of the settings above.” What setting is being talked about?

111: “To evaluate model calibration on common text understanding,” only understanding or also generation?

112: “Causal Language Modeling (CLM)” Subsequently, both the acronym and the full form are used.

115: designed facts generation task -> designed a a facts generation task

116: facts-related content. -> fact-related content.

120: the most possible one. -> the most probable one.

124: Model calibration improves with respect to both -> Model calibration improves with both

127: Improvement of calibration accuracy -> Improvement in calibration accuracy

128: facutality -> factuality

127: “Improvement of calibration accuracy is more significant on facutality or language understanding tasks than language modeling task.”
The introduction (lines 110-120) mentions three tasks to assess language comprehension, factuality, and reasoning skills. It would be useful to maintain consistency and clarify what is considered a task and what is considered an ability.

134: exacerbates such effect. -> exacerbates this effect.

135: Learning from reward model -> Learning from a reward model

142: “Calibration accuracy evolves consistently across different downstream tasks including factuality, language understanding or vanilla language modeling.” Same as for line 127.

150: hopefully can shed -> hopefully this can shed

151: trust-worthy -> trustworthy


Other comments:

-The design of Figure 1 could be improved.

-292: A short definition of the PILE dataset should be included.

-The captions in Figures 3 and 4 should be more explanatory. For example, the relationship between dataset and task should be mentioned.

-Figure 5 should include the accuracies.

---

> ### Author Rebuttal · Authors · 2023-08-29
>
> We greatly appreciate the reviewer's thorough examination and insightful comments. For missing references and suggestions on writing, we will try our best to improve in our next version. Following are some replies to your questions.
>
> ### **About the Limitation of Contribution**
>
> > There is no considerable contribution regarding SOTA ... The contribution of this work seems limited to analyzing the number of pre-training steps and different configurations of fine-tuning and RLHF, in addition to including CLM and fact generation tasks in the analysis.
>
> In this paper, the comprehensive examination of **SFT** and **RLHF** significantly broadens the scope of existing works by Kadavath et al., which already provides non-trivial contribution. These techniques are widely adopted in recent works or applications, consequently have a profound impact on a broad spectrum of researchers. Thus, it becomes crucial to deeply explore and address the implication of their effect on calibration behavior. In addition, we also evaluate popular efficient fine-tuning methods (e.g., LoRA) and demonstrate their regularization effects on calibration.
>
> Furthermore, we integrate the aforementioned aspects with the pretraining stage under a consistent evaluation protocol. The resulting conclusions cover the entire lifecycle of the LLM, providing a more comprehensive perspective on the emergence and evolution of calibration behavior.
>
> ### **Replies to questions**
>
> - **A. Does it refer to a understanding or generation?**
>   - It refers to generation task. Sorry for the words that lead to confusion.
>
> - **B. What were the criteria for selecting Reliability Diagram and ECE for the analysis? ...**
>   - For the selection of analysis tools, we follow most previous works to select Reliability Diagram and ECE. As you pointed out, such selection might be originally from Guo el al. (2017). We do cite this literature in our work, and will further clarify this in this part as well in our next version.
>
> - **C. The T-Rex dataset is mentioned to assess the Facts Generation task. However, PILE is also said to be used as a test (line 305). For what purpose is PILE used?**
>   - Here are the main reasons why we chose T-REx dataset and still kept the PILE. **Pile** covers a wide range of high-quality natural language texts ranging from web corpus to academic papers. It thus measures LM’s essential calibration behaviour under a very generalized distribution.
>   - By contrast, **T-Rex** is constructed from Wikipedia, and is more focused on wiki facts. And the results reflects how well-calibrated LM behaves in factual scenarios, where most of the miscalibration concerns exists.
>
> - **D. About inference time prompts.**
>
>   > In line 406 it says "As the models are fine-tuned on instructions in this stage, we add instruction prompt for all three tasks (see Table 1)". I understand that the models are fine-tuned only on Alpaca and OA. Are these prompts for running the inference on the tests?
>
>   - Yes, they are prompts for running the inference on the tests.
>
> - **E. About why Pythia is not used in alignment experiments**
>   - LLaMA is the most prevalent choices for existing alignment studies (e.g., Alpaca, Vicuna, WizardLM, etc.) , so it is selected to address the most common concerns in real application scenarios.
>   - Nonetheless, it indeed would provide more coherent analysis to also include Pythia in the entire alignment phase, and we will definitely add such experiments in our next revised version, e.g., training Pythia-1B with SFT and RLHF.
>
> - **F. About diversity of automatic and manual datasets**
>
>   > Regarding the comparison between Alpaca and OA datasets, it is pointed out that OA implies more calibrated models because of its greater diversity. In addition to ECE, do you have results from any other metrics that point to this fact? ...  Is it safe to say that the automatic methodology determines less diversity in instructions than manual methods?
>
>   - Besides ECE, we also demonstrate the distribution in semantic space of these two instruction-following datasets in **Figure 6**, we can see that Alpaca clearly exhibits more clustered pattern while OA is much more scattered.
>
>   - The discussion about diversity of different datasets is mostly based on intuition at present. Alpaca is constructed via in-context learning where LLMs generate new instruction-response pairs according to demonstration samples. While in manually annotated dataset OA, data are crowdsourced from annotators with different interests and backgrounds. For example, here are some instructions about emails in OA dataset. They display strong personal or emotional characteristics which are beyond the generation framework of Alpaca. By contrast, instructions in Alpaca are much more homogeneous.
>
>     ```python
>     # OA
>     [
>         "I do not want to work today. Can you imagine a disease which makes me sick for three days and formulate an email for my boss that I cannot work for the next days?",
>         "I'm completely overwhelmed with all the work I need to do. So many people expect stuff form me and I just can't handle it all. The more I work, the less I get done and I have so many unanswered email and things I promised to do, I'm scared to even go look at my inbox. What can I do?",
>         "write an email for me to a journal that i submitted a paper in it, but the review process is taking too long and the review tracking webpage did not change in a long time. I want to ask them what's wrong or to speed the process up."
>     ]
>
>     # Alpaca
>     [
>         "Create a response for a customer's email about a refund issue.",
>         "Generate an example of a formal email.",
>         "Write an email introducing yourself to a professor you will be taking an online course with."
>     ]
>     ```
>
> - **G. Missing accuracy results in alignment experiments**
>   - Sorry for the missing information. As the accuracy change is consistent with expectation and it is not the main target of our work, we did not present them in the paper initially. Here are the accuracies results of our alignment experiments (Results are splitted into two tables for better view). We will add them in the revised version appropriately.
>
>     |       | LLaMA | Alpaca_ep1 | Alpaca_ep2 | Alpaca_ep3 | OA_ep1 | OA_ep2 | OA_ep3 |
>     | :---: | :---: | :--------: | :--------: | :--------: | :----: | :----: | :----: |
>     | PILE  | 0.452 |   0.441    |   0.443    |   0.441    | 0.447  | 0.446  | 0.442  |
>     | T-REx | 0.407 |   0.403    |   0.407    |   0.406    |  0.41  | 0.412  | 0.411  |
>     | MMLU  | 0.366 |    0.336    |   0.354    |   0.372    | 0.331  | 0.318  | 0.334  |
>
>     |       | Alpaca_LoRA_ep1 | Alpaca_LoRA_ep2 | Alpaca_LoRA_ep3 | Alpaca_RLHF_ep1 | Alpaca_RLHF_ep2 | Alpaca_RLHF_ep3 |
>     | :---: | :-------------: | :-------------: | :-------------: | :-------------: | :-------------: | :-------------: |
>     | PILE  |      0.444      |      0.444      |      0.445      |      0.441      |      0.441      |      0.441      |
>     | T-REx |      0.415      |      0.409      |      0.411      |      0.405      |      0.405      |      0.405      |
>     | MMLU  |      0.362      |      0.367      |      0.369      |      0.37       |      0.368      |      0.369      |

---

### Meta-Review · Area_Chair_JC5N · 2023-09-18

**Recommendation:** 4

**Metareview:**

This paper performs an empirical analysis of the calibration of LLMs, after pretraining, instruction fine-tuning (full and LoRA), and RLHF, varying model size and training steps. Reviewers mostly agreed this was an important and informative analysis, though some of the findings (namely that larger models appear to be better calibrated after pretraining) are not new and replicate prior work. However, this work does provide analysis on additional datasets and training settings (after fine-tuning and RLHF). One reviewer questioned the soundness of evaluation of calibration using ECE, but seemed less concerned after the authors provided additional experiments. Overall this paper provides a comprehensive understanding of model calibration that goes beyond what was been studied previously, using existing accepted strategies for evaluating LLMs.

---

### Decision · Program_Chairs · 2023-10-07

**Decision:**

Accept-Findings

**Comment:**

This paper performs an empirical analysis of the calibration of LLMs, after pretraining, instruction fine-tuning (full and LoRA), and RLHF, varying model size and training steps. Reviewers mostly agreed this was an important and informative analysis, though some of the findings (namely that larger models appear to be better calibrated after pretraining) are not new and replicate prior work. However, this work does provide analysis on additional datasets and training settings (after fine-tuning and RLHF). One reviewer questioned the soundness of evaluation of calibration using ECE, but seemed less concerned after the authors provided additional experiments. Overall this paper provides a comprehensive understanding of model calibration that goes beyond what was been studied previously, using existing accepted strategies for evaluating LLMs.